# Coarse-Grained Simulations on Polyethylene Crystal Network Formation and Microstructure Analysis

**DOI:** 10.3390/polym16071007

**Published:** 2024-04-07

**Authors:** Mohammed Althaf Hussain, Takashi Yamamoto, Syed Farooq Adil, Shigeru Yao

**Affiliations:** 1Central Research Institute, Fukuoka University, Fukuoka 814-0180, Japan; 2Graduate School of Science and Engineering, Yamaguchi University, Yamaguchi 753-8512, Japan; 3Department of Chemistry, College of Science, King Saud University, P.O. Box 2455, Riyadh 11451, Saudi Arabia

**Keywords:** MD simulations, semi-crystalline structure, polymer physics, polymer network, crystallization, relaxation, entanglements, mechanical properties

## Abstract

Understanding and characterizing semi-crystalline models with crystalline and amorphous segments is crucial for industrial applications. A coarse-grained molecular dynamics (CGMD) simulations study probed the crystal network formation in high-density polyethylene (HDPE) from melt, and shed light on tensile properties for microstructure analysis. Modified Paul–Yoon–Smith (PYS/R) forcefield parameters are used to compute the interatomic forces among the PE chains. The isothermal crystallization at 300 K and 1 atm predicts the multi-nucleus crystal growth; moreover, the lamellar crystal stems and amorphous region are alternatively oriented. A one-dimensional density distribution along the alternative lamellar stems further confirms the ordering of the lamellar-stack orientation. Using this plastic model preparation approach, the semi-crystalline model density ($\rho_{cr}$) of ca. 0.913 g·cm^−3^ and amorphous model density ($\rho_{am}$) of ca. 0.856 g·cm^−3^ are obtained. Furthermore, the ratio of $\rho_{cr}/\rho_{am}$ ≈ 1.06 is in good agreement with computational (≈1.096) and experimental (≈1.14) data, ensuring the reliability of the simulations. The degree of crystallinity ($\chi_{c}$) of the model is ca. 52% at 300 K. Nevertheless, there is a gradual increase in crystallinity over the specified time, indicating the alignment of the lamellar stems during crystallization. The characteristic stress–strain curve mimicking tensile tests along the z-axis orientation exhibits a reversible sharp elastic regime, tensile strength at yield ca. 100 MPa, and a non-reversible tensile strength at break of 350%. The cavitation mechanism embraces the alignment of lamellar stems along the deformation axis. The study highlights an explanatory model of crystal network formation for the PE model using a PYS/R forcefield, and it produces a microstructure with ordered lamellar and amorphous segments with robust mechanical properties, which aids in predicting the microstructure–mechanical property relationships in plastics under applied forces.

## 1. Introduction

Semi-crystalline polymers have diverse applications in society and are integral to daily life [1,2]. Owing to their dual-phase structural complexity, comprising highly ordered crystalline lamella molecules and randomly arranged amorphous molecules, they foster both elastic and plastic deformations within the plastics. It is challenging to achieve a balance between the two structural phases. However, understanding them comprehensively at the molecular level [3,4,5] could leverage designing new materials or predicting the optimized condition for their processing. Controlling the molecular arrangements of semi-crystalline materials could enhance mechanical strength, durability, and thermal stability and challenge the industry’s conventional products, from packaging to biomedical applications [6,7,8,9]. The lamellar staking structure in semi-crystalline materials is essential in determining the thermal properties, anisotropic behaviors, and mechanical properties such as tensile strength, stiffness, toughness, and ductility [3,4,9]. A well-oriented crystalline lamellar polymer chain in plastics improves tensile strength and stiffness by resisting deformations due to its tightly packed crystal arrangements. On the other hand, randomly arranged amorphous molecules exhibit toughness and ductility, in which the chain arrangement facilitates plastic deformation. A cursory look at the literature shows that the experimental polymer processing conditions have not successfully controlled the crystalline lamellar orientation in the desired direction, which could be attributed to the heterogeneous distribution of the entanglement density that directly affects the crystallization from the melt [10,11,12,13,14,15,16]. However, it is evidenced from oscillatory shear deformation studies that lamellar crystal growth can be controlled in plastics by tuning the process conditions, which enhances the mechanical strength of the materials by a thousand-fold [17,18,19]. Additionally, the static shear deformation proved that lamellar crystal axis (C-axis) growth occurs perpendicular to the direction of deformation [20,21,22]. Using such approaches, it could be concluded that processing techniques are crucial in shaping industrial products. However, producing such experimental results with accuracy and precision is time-consuming and challenging, and most often, such attempts are futile, which costs time and money.

MD simulations, on the other hand, are crucial in comprehensively understanding polymer chain dynamics and physical state transformation over the specified time scales [23]. The qualitative information obtained in MD simulations outperforms the arguable quantitative time scales, more importantly when the macromolecular dynamics and insights are targeted from nanoseconds (ns) to microseconds (µs) and sometimes to milliseconds (ms) length simulations—compared with seconds (s) to minutes (m) in experiments. However, there are limitations with MD techniques, such as forcefield accuracy and approximations, time and length scale, and sampling and conformation space; MD simulation techniques are still highly regarded for microstructure–property prediction under various conditions [3,4,23]. Numerous MD simulations have been reported with the induced crystallization of HDPE with well-oriented lamellar in alternative arrangements near room temperature [24,25,26,27,28]. In a recent study of isothermal crystallization, the authors probed the preparation and characterization of alternating lamellar HDPE models using the Transferable Potentials for Phase Equilibria (TraPPE) forcefield [3]. This study justifies the reliability of the isothermally crystallized cubic and rectangular cell box models at 300 K by their characteristic properties, such as density, $\chi_{c}$ and, mechanical properties. However, the details of microstructure recrystallization and the mechanical properties of two-layered lamellar orientation models are not compared, which could add more information to the molecular behavior and chain dynamics of plastics. A vital aspect of the chain dynamics, the crystal network formation in polymer crystallization, and the mechanical properties are investigated at 300 K for the 10-chain PE model. It is known from the above study that the 10C_1000_ model is the most suitable model for evaluating the application of plastics at the molecular level. 

As the polymers are the macromolecules, CGMD simulations are recommended over atomistic simulations to obtain fast and reliable results [26]. Though the CGMD simulations are practical in computation, they miss vital molecular details such as precision in bond dynamics, electrostatic interactions, and phase transformations. On the other hand, plastic deformation-dependent models have been widely studied, and the thermal and mechanical properties have been evaluated for both amorphous and semi-crystalline models to date [3,4,27,28,29,30,31,32,33,34]. However, the literature information is insufficient and demands novel optimized strategies and methods without negotiating fast and reliable outcomes. From the author’s perspective, gaining prior information from the CGMD simulations is the best strategy to understand and control the lamellar orientation at the molecular level, since the CGMD tools are the most effective for studying plastic’s complex structure (semi-crystalline) behavior [23]. Eventually, it prevents the time and cost involved in experimental attempts, which proceed based on assumptions. 

The present study uses uniquely packed lamellar stems to prepare the united atom (UA) forcefield-based polyethylene semi-crystalline model (the simplest possible form of coarse-grained models). It explores the model’s reliability in obtaining the physical and mechanical properties under applied external forces. In this simulation, the PYS/R forcefield opts for recrystallization, which is highly regarded for PE computation and relatively produces the results seen for the Transferable Potentials for Phase Equilibria (TraPPE) forcefield [35,36]. Generally, Rutledge et al.’s models extensively predict the mechanical properties using the PYS/R forcefields [37,38]. We adopt this highly optimized forcefield for our model building to uncover the crystallization kinetics, while obtaining spontaneous (with MD tools) crystal nucleation at the molecular level, and justify its microstructure characteristics by evaluating the mechanical behavior under uniaxial deformation at 300 K [39].

## 2. Simulation Methodology

### 2.1. Preparation and Packing of Multi-Folded HDPE Chains

The initial HDPE-folded chain model is prepared similarly to the previously published work by the same group [3]. The model preparation for this study from a single chain to the 10 chains containing the initial model is shown in Figure 1A–D. An initial HDPE polymer chain model having 1000 methylene units (CH_2_) is prepared in a cubic box dimension of 27.0 Å using the Enhanced Monte Carlo (EMC) tool (Figure 1A) [40]. Each CH_2_ unit is represented as a united atom (UA) in 1000 repeating units of PE chain (C_1000_), and it is obtained at room temperature (RT; 300 K) and density (ρ) of 0.85 g·cm^−3^. Each UA molecular weight is 14.02 amu. Individually, the isolated C_1000_ model chains are simulated in Material Studios’ Forcite module [41] to melt and fold them (Figure 1B). The canonical ensemble (NVT) is used to melt and fold the chains at 473K using the Berendsen thermostat [42]. These two independently simulated chain conformations have loop/fold and tail segments of semi-crystalline structures [5,43,44]. As shown in Figure 1C, a rectangular cell dimensional box with a lamellar-stacked oriented model is manually created using the Material Studios’ visualizer, and the total energy of the simulation box is minimized at 300 K using a ‘smart’ algorithm. The 10-chain packed simulation box lengths in Å are 50.0, 50.0, and 100.0 (Appendix A). Our recent paper discusses thoroughly the initial model’s details [3]. Eventually, the msi2lmp [45] tool is used to convert the Materials Studio files to the Large-scale Atomic/Molecular Massively Parallel Simulator (LAMMPS) [46] data files for standard MD simulations using Paul–Yoon–Smith (PYS) forcefield parameters modified by Rutledge et al. (PYS/R) [37,38,39]. A description of PYS/R forcefield parameters mimicking the interatomic interactions among the PE chains is shown below. 

Table 1 forcefield parameter units accurately represent the molecular interactions and their behavior: force constant (*K_b_*) for bond potential in kcal/mol/Å^2^; force constant for angle potential (*K_θ_*) in kcal/mol/radian^2^; dihedral potential (*C_0_*) in radians; equilibrium angle (*θ_eq_*) in degrees; the equilibrium bond length (*r_eq_*), range of the potential (*σ*), and cut-off distance (*r_cut_*) are in Å; the empirical parameters for the first harmonic term (*C*_1_), amplitudes of the first (*C*_2_) and second cosine term (C_3_), and strength (*ε*) are represented in kcal/mol. The bonded and non-bonded interactions are computed using Equations (1)–(4).
(1)$$Bond\;stretching\;\sum_{bonds}K_{b}{\left(r-r_{eq}\right)}^{2}$$
(2)$$Angle\;stretching\;\sum_{angles}K_{\theta }{\left(\theta -\theta_{eq}\right)}^{2}$$
(3)$$Dihedrals\;\;\;\;\;\;\;\;\;\;\;\;\;\;\;\;\;\sum_{dihedral}C_{i}\left({cos}^{i}\left(\phi \right)\right)$$
(4)$$\mathrm{Lennard}\text{-}\mathrm{Jones}\;(\mathrm{van}\;\mathrm{der}\;\mathrm{Waals})\;\sum_{i}\sum_{j\neq i}{4\varepsilon }_{ij}\left[{\left(\frac{\sigma_{ij}}{r_{ij}}\right)}^{12}-{\left(\frac{\sigma_{ij}}{r_{ij}}\right)}^{6}\right];\;\sigma_{ij}=\frac{\sigma_{ii}+\sigma_{jj}}{2};\;\varepsilon_{ij}=\sqrt{\varepsilon_{ii}\varepsilon_{jj}}$$

The interaction parameters between the atomic units *i* and *j* are computed using the Lorentz–Berthelot mixing rules.

### 2.2. Melting, Equilibration, and Isothermal Crystallization

The 10C_1000_ chain model is energy-minimized using a conjugate gradient algorithm in LAMMPS with a time step of 1fs. The microcanonical ensemble (NVE) minimizes the energy in 5 × 10^5^ steps by supplying the random velocities at 300 K. To obtain the isotropic melting model at 450 K, the energy-minimized model is heated at 450 K for 1 ns in the subsequent step using the NVT. This is followed by the equilibration of the heated model at 450 K and 1 atm using the isothermal–isobaric ensemble (NPT) conditions for 10 ns time length to ensure that the T, P, density (*ρ*), and the potential energy (PE) are entirely converged. To further ensure the energy parameters, the decomposition of the energy components is also tested, as shown in Appendix A. The PE energy fragments, such as bond, angle, dihedral, and van der Waals energies, are also computed at an interval recording of 10 ps for 10 ns. Note that ∆t is 2fs, which is applied to simulate the melting and equilibrations, and the same is used to compute the quenching and isothermal crystallization. The equilibrated models are quenched in two steps: in the beginning, the 450 K model is cooled down gradually to 400 K at a cooling rate of 10 K/1ns, and in the subsequent step, it is slowly cooled down to 300 K at a cooling rate of 5 K/1ns. 

### 2.3. Analysis of the Model

#### 2.3.1. One-Dimensional Density

The semi-crystallized structure segment orientations in the simulation box along the one-dimensional density profile are calculated using the chunk (bin) option available in LAMMPS. The mass density along the z-axis is computed using the mass density/volume (in g·cm^−3^) of each slab spatial bin/chunk size of 0.5 Å. A total of 10 sample averages are considered for the density distribution along the z-axis using an NVT ensemble at 300 K while reporting the data of 10 averages.

The ratio of the number of atoms located within each chunk ($N_{i}$), and volume of each chunk ${(V}_{i})$, results in the density along the z-axis {$\rho \left(z\right)$}, as shown in Equation (5).
(5)$$\rho \left(z\right)=N_{i}/V_{i}$$

#### 2.3.2. Degree of Crystallinity ($\chi_{c}$)

The phase change from molten to crystalline state during the isothermal crystallization is measured by computing the $\chi_{c}$ using the local orientation parameter. For the second-order Legendre polynomial ($P_{2})$ calculations, the coarse-grained molecular dynamics program (COGNAC version 8.4) in the open computational tool for advanced materials technology (OCTA) [47] is applied using in-house-built Python code. The following Herman’s orientation factor (Equation (6)) is employed to calculate the degree of crystalline order $P_{2}\left(r\right)$ immediately after measuring the mid-point of two adjacent CH_2_-CH_2_ UA bonds; this mid-point is the ith UA located at the position r_i_.
(6)$$P_{2}\left(r\right)=\langle 3{cos}^{2}\theta_{i,j}-1\rangle /2$$

The angle (*θ_i,j_*) is that between chord vectors bi and bj within the same mesh-cell size of 8Å located at the representative position r. The average is taken over all pairs of chord vectors within the mesh cell. The estimation of crystallinity ($\chi_{c}^{mesh}$) comes from the number of mesh-cells Nc that have more local order parameters than a 0.4 threshold cut-off, divided by the total number of mesh-cells in the system N_total_. The cut-off of 0.4 is selected based on the recommendation of previous publications [48].
(7)$$\chi_{c}^{mesh}=N_{c}(P_{2}>0.4)/N_{total}$$

Similar strategies have been observed in the order parameters and $\chi_{c}\;$calculations of previous publications [3,4,27,28].

#### 2.3.3. Mechanical Properties

Semi-crystalline structures, in general, are anisotropic in their morphology, and it would be necessary to examine the tensile test in the simulation box’s x, y, and z directions. Herein, the semi-crystalline models at 300 K and 0 atm are uniaxially deformed in the transverse direction at a strain rate ($\dot{\varepsilon }$) of 10^10^ s^−1^. The strain rate is equivalent to the strain rates used for previous HDPE amorphous [49] and crystalline models [3] and uniaxially deformed to 1000% of the simulation box in 1 ns time length. To scan the microstructure characteristics in detail, the ∆t is reduced to 1fs. The intrinsic property of polymeric materials is their tensile strength; the stress–strain curves are drawn for all three directions of deformations by recording the MD simulation trajectories at interval of 1 ps. The average data for the three sample calculations are shown. The moving averages of the pressure tensors along the x (p_xx_), y (p_yy_), and z (p_zz_) directions are shown in the S-S curve.

The strain rate ($\dot{\varepsilon }$) is computed using Equations (8) and (9), which represents the change in uniaxial deformation strain (ε) with a formation strain (∆ε) with change in time (∆t).
(8)$$\dot{\varepsilon }=(∆\varepsilon /∆t)$$
(9)$$\varepsilon =(\left(L-{\;L}_{0}\right)/L_{0})=(∆L/L_{0})$$

*L* and $L_{0}$ are the final and initial lengths of the box along the deformation axis of the simulation box, and measured using the Henchy strain. 

All the simulations are performed in three-dimensional space using the Nosé–Hoover thermostat and barostat [50] to control the temperature and pressure. During the simulations, the relaxation time is adjusted after every 100 timesteps, and the minimum distance between neighboring particles is binned 0.4 times. The linear momentum of all atoms in MD is equally used to control the movements of the particles that stabilize the MD simulations. For the post-processing, to measure the χc in the studied models, the OCTA version 8.4 [47] is used. For primitive path analysis and to account for the average number of entanglements per chain (<Z>), Kröger’s *Z1+* code is performed on the amorphous and crystalline states [51,52]. The open visualization tool (OVITO) [53] Pro version 3.10.1 is used for the graphical representation of the models.

## 3. Results and Discussion

### 3.1. Melt-Equilibration and Deep Quenching

Understanding the isothermal crystallization phenomenon in PE model-making from classical MD simulations is computationally challenging due to the long chain relaxation and ordering into a perfectly crystalline orientation. Fortunately, several PE models possessing varying lengths for the polymer chains and sizes have been explored at the molecular level by addressing the crystallization kinetics and microstructure characteristics, such as entanglements analysis [36,54]. However, to obtain a stabilized model at atmospheric conditions using a reasonable time and limited computational resources, the interatomic forcefield parameters and initial chain packing of the polymer model play a crucial role, and its effect on the simulations needs to be tested at each stage of the simulations. Thus, in our MD simulation standard protocols, the steady state formation of the molten model is elucidated before proceeding to a production run for isothermal crystallization. The PYS/R forcefield-dependent thermodynamic parametric data in Figure 2 illustrate the formation of a stable melt structure. The parametric data are listed below and obtained by calculating density, temperature, pressure, PE, and its energy components at 450 K and 1 atm pressure conditions using NPT ensemble averages. Before that, to heat the model from 300 K to 450 K, the canonical ensemble (NVT) simulation for 1 ns is performed and included in the above parameters of Figure 2. 

The results in Figure 2, from A-D with steady flow over the 10 ns MD simulations at 450 K and 1 atm, suggest that the model has reached the anticipated steady state and the 10 ns time length for the isotropic mixture formation is successful. The PE components’ stabilization for the same conditions is shown in Appendix A. The PE components are the additional measure of the interatomic interaction changes of the microstructure. The bond, angle, dihedral angle, and van der Waals components move in a relatively stable manner within a 2 ns time length, and no sudden drift is observed to deny the formation of an isotropic mixture of the multiple chains, which initially started as independent and isolated. The parameters for NPT melt equilibration reach the steady state, and the fluctuations are within the acceptable range, confirming that the results support steady state formation. In addition, there are stable PE coiled structure similarities at the processing temperature, i.e., 450 K, which usually appear in the optimized conditions for PE processing.

To cool down the isotropic melt structure, temperature quenching from 450 K to 300 K and 1 atm condition using NPT ensemble is performed in two stages: a high quenching rate of 10 K/1 ns from 450 K to 400 K, and a slightly slower quenching rate of 5 K/1 ns from 400 K to 300 K. The variation in the quenching rates is because fast cooling could lead to glassy-state formation in plastics; this was observed as well in previous reports, where a much slower temperature-quenching protocol was adopted for crystallization [55,56]. Slower quenching rates are adopted in the simulations to avoid such phase change from the molten state and prepare the desired plastic PE model. The phase change during the quenching with different temperature cooling is reported by computing the total density, and to account for the compatibility of the polymer chains from the coiled state to the much denser globular amorphous state, the average number of entanglements per chain is computed as shown in Figure 3.

Figure 3 demonstrates a linear density growth for the 10C_1000_ model in the temperature-quenching stage, indicating that the random chains are aligned in the simulation box to reorient and organize themselves. The initial and final densities of the models are marginally overestimated with distinct input structures; the physical state still represents the corresponding temperature-holding structure. For example, the 10C_1000_ model at 450 K shows a coiled and unentangled structure density of ca. 0.759 g·cm^−3^, close to the experimental PE density of 0.760 g·cm^−3^ [36]. After temperature quenching to 300 K, the same exhibits a dense polymeric spherical structure with a reduced density of 0.865 g·cm^−3^ and is slightly higher than the experimental PE density of 0.85 g·cm^−3^ at 300 K [31,36,57,58,59,60]. The lowering of the density indicates that the free-coiled structure is shrinking in size, which further suggests that the polymer chains are compressing due to the loss of thermal motion holding from the melting temperature. This is also supported by the average number of entanglements per polymer chain, which was initially noted as 4.0. After the quenching increases due to the topological entanglement, the average number of entanglements per polymer chain is 7.6. From Figure 3B, it can be realized that entanglements show a gradual increase in the models as the temperature is quenched from 450 K to 300 K. Consequently, from the above parameter metrics, it is understood that the quenching is done much more slowly in the MD protocol to form a realistic polymer model. This quenching strategy prevents the formation of a glassy state, a commonly observed phenomenon in polymer crystallization at the molecular level.

### 3.2. Isothermal Crystallization from Melt

Isothermal crystallization of the 10C_1000_ model is carried out for the 1 µs time length to ensure the formation of semi-crystalline models with, or close to, the realistic total densities of ca. 0.90–0.940 g·cm^−3^ at 300 K and 1 atm using NPT ensembles. Figure 4 illustrates the model transformation from its beginning input data (Figure 4A), its 450 K NVT-melting and NPT-equilibration model (Figure 4B), and step-wise intermediate quenching steps at 400 K (Figure 4C), at 350 K (Figure 4D), at 300 K (Figure 4E), and finally the isothermally crystallized model at 300 K (Figure 4F). Figure 4F confirms that a semi-crystalline HDPE model with a density of ca. 0.913 g·cm^−3^ is prepared using PYS/R forcefield parameters at 300 K and 1 atm pressure conditions. The crystallization also involves the phase transformation of the amorphous state to the semi-crystalline state as the MD simulations progress from 0 ns to 1 µs at 300 K. 

The starting input 10-chain structure with a folded chain conformation is converted into semi-crystalline models with stacked lamellar orientations. The model’s densities are amendable to the experimental densities at an equilibration state of 450 K (0.759 g·cm^−3^), quenched state densities of 0.856 gm.cm^−3^ at 300 K, and isothermal crystallization temperature density of 0.913 g·cm^−3^ at 300 K. However, a discrepancy is observed in the density at 450 K and 300 K after quenching, where the total density is slightly overestimated, while due to the slow rate of chain alignment at the isothermal crystallization at 300 K, the computed density is underestimated. The characteristic densities of the amorphous state ($\rho_{am}$), 0.856 g·cm^−3^, and the crystalline state density ($\rho_{cr}$), 0.913 g·cm^−3^, are obtained. Specifically, the $\rho_{cr}$ is underestimated compared to the experimental density of 0.97 g·cm^−3^. Nevertheless, the simulated model’s ratio $\rho_{cr}/\rho_{am}$ is ~1.06, which is marginally close to the experimental value of 1.14 [61] and computational value of 1.096 [62,63], evident that the calculations are marginally in good agreement with the ideal models in the literature. 

The quantitative density measurement in Figure 5A illustrates a gradual increase in density during isothermal crystallization. This indicates that the chain straightening and alignment of the lamellar stems over time occurs due to van der Waals interactions among the PE chains, consistent with a familiar molecular arrangements concept reported for PE crystallization [36]. 

A multi-crystal nucleus formation in the isothermal crystallization evidences the anisotropic nature of the morphology and the heterogeneous crystallization. This is also supported by $\chi_{c}$ graphs in Figure 5B, which show that the crystal growth is linear. Figure 5A–C prove that the crystal growth seems natural. In the process of isothermal crystallization at 300 K, the seeds of the crystal nucleus are observed at the beginning of Figure 5C, indicating that the prior crystallization is obtained in the quenched state temperature of 300 K. This arises due to the quenching protocol that is applied in the simulations. Ultimately, a small $\chi_{c}$ for the 10C_1000_ model is 3.8%, a noticeable measurement. As the isothermal crystallization progresses, faster chain alignments occur, increasing crystal lamellar thickness. Specifically, after the 200 ns time length and until 800 ns MD simulations time lengths, the crystallization is accelerated, which can be easily observed in Figure 5B,C. Finally, $\chi_{c}$ of 52% is achieved for the semi-crystalline model, which is relatively lower than the experimental crystallinity reported for the HDPE at room temperature. 

Although the lamellar texture size and distributions are heterogeneous, the high-level ordering of the lamellar stems as a part of the crystalline region is obtained even in PYS/R forcefield simulation models. In contrast, the TraPPE forcefield model ordering is highly heterogeneous, even with a lower $\chi_{c}$ for the 10C_1000_ model preparation [3]. This concludes that the PYS/R forcefield generates PE semi-crystalline models with similar structural and property features as seen for TraPPE forcefield models at 300 K and 1 atm conditions using a similar MD protocol.

Figure 4F illustrates polymer chain orientation in box and unwrapped box forms. The chains in crystalline order and amorphous forms represent alternative forms. However, the size of the two lamellar segments is different, and a small portion of the amorphous region exists between these two lamellar orientations, as was seen in the previous results of PE semi-crystalline models [3,31,54,57,58]. The characteristic crystal network bridge/tie, loops, and tails chain forms are also obtained, indicating the microstructure has all the necessary HDPE structural characteristics with the known densities at ambient conditions. These results support the previous results of the HDPE semi-crystalline models [57] at 300 K and 1 atm pressure conditions, an ideal model for studying polymer applications such as structural deformations for engineering applications. From this, the structure–property relationship can be predicted.

### 3.3. Z-Axis Density {ρ(z)}

The spatial distribution of mass offers a detailed view of the arrangement of molecules within the polymer model. It helps distinguish between the randomly oriented amorphous and highly ordered crystalline states. The Z-axis dimensional density in the obtained model is evaluated using the small bins moving along the same direction (Section 2.3.1). The computed results for Z-axis density in the 10C_1000_ model are shown in Figure 6, which indicates the variation in the density in small bins along the lamellar stack orientation formed. It can be quickly confirmed from the structure and the density profile along the Z-axis that the lamellar stack orientation containing the semi-crystalline model is obtained using this approach. It should also be noted that the lamellar stem sizes and amorphous regions in the prepared models are uncontrollable and could vary in size and spatial distribution. The total density is 0.913 g·cm^−3^, and the one-dimensional density varies from ca. 0.9–1.0 g·cm^−3^ for the crystalline state and 0.85–0.89 g·cm^−3^ for the amorphous state. The amorphous and crystalline parts’ densities fluctuate slightly while moving along an axis, indicating that these two regions have trapped the orientation of their rival atom. This is commonly seen in the MD simulations of the reported models, where the densities fluctuate while crossing from one physical state to another [54]. However, the lamellar stems differ in length and size along the same axis. Quantitative measurement of the lamellar length along the z-axis orientation is shown in Figure 6: at the beginning of the cell, the length is approximately 38 Å, while on the extreme right-hand side of the cell, the lamellar stem length is 17 Å. In comparison, the middle part of the cell accountable for the amorphous region has a region of ~45 Å. This implies that the lamellar stems are not perfectly aligned, as commonly studied in Rutledge et al.’s in-house build semi-crystalline models [16,30,31,32] and similar models which produced the same result [27,43,54,59].

It is evident that the 10C_1000_ model is prepared from the MD simulations by following an isothermal crystallization strategy, and preparing a perfectly aligned Rutledge et al. lamellar stack model is impossible for non-experts of the field and quite challenging for the simulation experts. However, the authors have prepared the models from 1 µs time length because the models are highly ordered crystalline structures, which are still less than the experimental densities but could serve as better models for predicting the actual behavior of HDPE materials. However, a small attempt by the authors paved the way for the acquisition of such PE models in a short time, which is the secondary aim of the study, and it is achieved for the 10C_1000_ model.

### 3.4. Mechanical Properties and Microstructure 

The elastic modulus and tensile strengths are the intrinsic mechanical properties of polymers that are mostly dependent on the direction of the lamellar segments. The semi-crystalline 10C_1000_ model has lamellar orientation along the z-axis and no orientation along the x- or y-axis direction. To gauge the mechanical behavior along the x-, y-, and z-axis directions, the uniaxial deformation in all directions is performed on the 10C_1000_ model, in which the microstructure network includes bridge, loops, and tails that are formed in the MD simulations at the 300 K and 1 atm conditions using the NPT ensemble (Figure 4). Figure 7 illustrates S-S curves along the x, y, and z-axis directional deformation for the 10C_1000_ semi-crystalline model at 300 K and zero-pressure conditions. The S-S curve behavior along the z-axis direction is evidence of the semi-crystalline model with an alternative lamellar segment orientation. Furthermore, the upper and lower parts of the crystalline regions show different degrees of orientation, representing a heterogeneity of the lamellar thickness. Nevertheless, the model with distinct lamellar thickness along the z-axis, the alternative arrangement, is in line with the models mentioned earlier in the literature; the popularly well-known PE plastic model by Rutledge et al., is mainly suitable to study plastic behavior under external forces to unravel the microstructure–property relationships [16,31,40,57,58]. Similar to the previous model studied by the authors using the TraPPE forcefield [3], the 10C_1000_ model is slightly different in crystalline orientation along the z-axis direction. Meanwhile, the alternative lamellar orientation along the z-axis is still maintained irrespective of the choice of the forcefield, indicating that the forcefield change affects the degree of crystallization and the thickness of the lamellar in the microstructure [31,58]. 

The elastic modulus of the models along the z-axis direction is higher than that of the x- and y-directions due to the alternative lamellar orientation along the covalent bonds of the polymer chain axis in z-axis deformation. This is the main reason for the sharp elastic regime stress tensors in Figure 7C, coming initially from restoring forces of the polymer segments followed by the permanent change/plasticity and, finally, the stretching of the covalent bonds along the strain deformation. On the other hand, the x- and y-axis directions’ pressure/stress tensors in Figure 7A,B come from the weak van der Waals forces, due to which a long plastic regime is observed in the elongation, with a much more bent elastic regime than the above-mentioned deformation direction viz. z-axis. 

The S-S curves analysis in Figure 7A,B support the non-alternative arrangements for lamellar segments in the microstructure; principally, the weak van der Waals forces resist the x- and y-axis directions’ pressure/stress tensors, so instead of a long plastic regime in the elongation, a steady elastic region as like Figure 7C is observed, supporting the anisotropic morphology of the models. The elastic modulus of the models along the z-axis direction is higher than that of the x- and y-directions due to the alternative lamellar orientation along the covalent bonds of the polymer chain axis in z-axis deformation, a characteristic S-S curve at ambient conditions for HDPE.

The intrinsic mechanical properties of the 10-chain model comprise a reversible elastic regime and a yield point (which appeared only in Figure 7C), followed by a non-reversible deformation plastic regime. A steady applied stress plastic regime behavior mainly appeared in Figure 7A,B, confirming the absence of lamellar orientation along the x- and y-axis in the simulation box. Previous reports on the alternative lamellar and amorphous model building and its mechanical properties are well documented using a range of forcefield parameters [31,36,40,57,58,59]. Specifically, consistent with our study, the PYS/R forcefield parameters for simulating mechanical properties of the in-house HDPE models, which are prepared by combining the pre-crystalline arrangements in a cell and randomly oriented amorphous regions using Monte Carlo simulations, are widely explored [59], and the molecular-network-dependent mechanical properties are also quantitatively revealed for similar structure models [44,57,59]. Note that the tie/bridge, fold, and tail networks are manually determined during the sample preparation. A significant role played by the bridge or tie molecules for the ductile nature in the HDPE models is seen in this study after the ~90% pre-fracture to the ~350% plastic deformation region, which is attributed to the bridge chain structure contributing to mechanical stress. The chain slip is also seen during the plastic deformation, followed by the crystal slip slightly before the ultimate tensile strength near 350% of deformation. It is also seen in the microstructure behavior as the deformation progresses; the cavitation mechanism appears near the 90% deformation in the z-axis—which is absent for x- and y-axis deformations. The bridge (tie) molecules are principally responsible for the ductile nature of the model, which is formed in the isothermal crystallization using Paul–Yoon–Smith, a version modified by Rutledge et al. (PYS/R) forcefield parameters.

To assess the sensitivity of the prepared model towards the applied strain rates, we measured two additional S-S curves for the strain rates, i.e., 10^9^ and 10^11^ s^−1^, using the same conditions and ensemble, as shown in Figure 8. It could be observed that the S-S curve behavior is sensitive to the change in the strain rate, which appears to be polymeric amorphous in the case of the low strain rates at 10^9^ s^−1^. The S-S curve is more ductile, indicating that the lamellar crystals are exposed to low strain levels and the restoring forces are allowing the lamellar crystals to integrate. On the other hand, the 10^11^ strain rate highly deformed the polymeric chains, and the applied forces have high strength over the resisting forces, due to which the mechanical stress is almost near 0 and steady along the 1000% deformation of the simulation box. So, the chosen $\dot{\varepsilon }$ (10^10^ s^−1^) is primarily suitable for the model under consideration, whereas it is highly sensitive to the change in the small strain levels. This supports that the choice of the strain rate is crucial in obtaining realistic mechanical behavior of plastics.

Appendix A also studies and reports structural deformations in the amorphous state along the three axes. Unlike the semi-crystalline model deformation for 1000% deformation, the simulation time for deformation, the amorphous models at 300 K and zero-pressure conditions are only deformed up to 500% of the simulation box. The S-S curve has shown an amorphous HDPE nature in mechanical behavior in all directions. This indicates that the amorphous state has its characteristic S-S behavior (a similar example of amorphous HDPE models). 

The molecular deformation mechanism involving alignment and stretching of polymer chains followed by a pre-cavitation in the uniaxial deformation along the z-axis is evidence that the lamellar orientation is parallel, and they are alternatively arranged in the preparation of the model, which is obviously needed in HDPE tensile tests. A distinct mechanical behavior is seen along the x, y, and z-axis directions, indicating an anisotropic nature. Such anisotropic nature is attributed to the heterogeneous morphology; the lamellar and amorphous segments have different size distributions and natural amorphous and crystalline segment orientations in experiments. The S-S curve along the z-axis direction suggests that the lamellar orientations are uni-directional, and the covalent bonds (C-C; UA) mainly resist the stress tensors in the plastic deformations.

## 4. Conclusions

A comprehensive procedure to obtain a PE semi-crystalline model at 300 K and 1 atm is investigated using the standard PYS/R forcefield. The microstructure–mechanical properties are evaluated at different strain rates to justify their reliability for plastic applications. Isothermal crystallization of the HDPE model from melt produces a microstructure with alternate lamellar and amorphous orientations. The density distribution along the alternate direction of the plate orientation axis further confirms the orientational ordering of alternate lamellar and amorphous regions [49,60]. The two-order parameter calculations demonstrate the gradual increase in crystallinity in isothermal crystallization, and the semi-crystalline structure is obtained with a degree of crystallinity of 52% [3,4]. A close ratio of simulated $\rho_{cr}/\rho_{am}$ ≈ 1.06 and experimental (≈1.14) [61] is observed, which is in line with the simulation results for PE models of ≈ 1.096 [62,63]. The cavitation mechanism involves the alignment of lamellar rods along the deformation axis exhibiting a stiffer elastic regime followed by tensile strength at yield ca. 100 MPa. A pre-fracture near 90% strain loading, followed by a 350% tensile strength at the break due to the covalent bond stretching of tie chains, apparently supports the typical S-S curve of HDPE [31,58,59]. In general, this study emphasizes the formation of a crystalline network for the PE model using the PYS/R forcefield, and it produces a microstructure with ordered lamellar and amorphous segments with robust mechanical properties that are consistent with previous results [44,57,59]. This study guidelines obtaining fast but reliable PE semi-crystalline models to predict the structure–property relationship for industrial applications. 

## Figures and Tables

**Figure 1 polymers-16-01007-f001:**
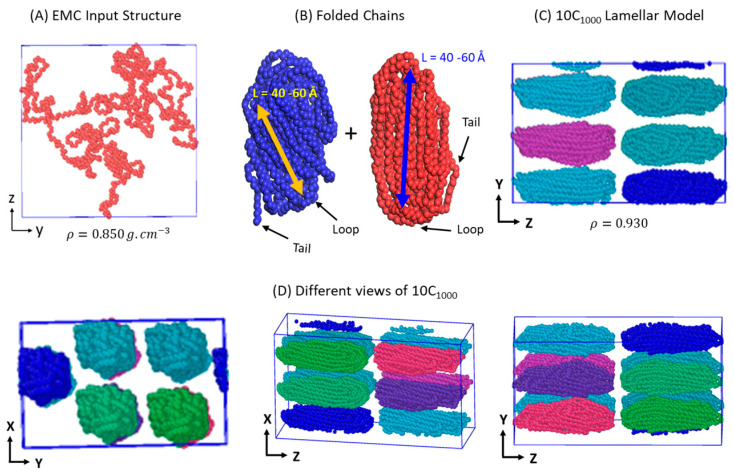
Pictorial representation of the initial models: (**A**) Polyethylene chain with 1000 UA; (**B**) Independent chains folding in Material Studio 2022 (version 22.1.0.3462, Accelrys, San Diego, CA, USA) Forcite module; (**C**) Folded chains with the penetrating interphases; (**D**) Folded chains packed to obtain the two-lamellae layers.

**Figure 2 polymers-16-01007-f002:**
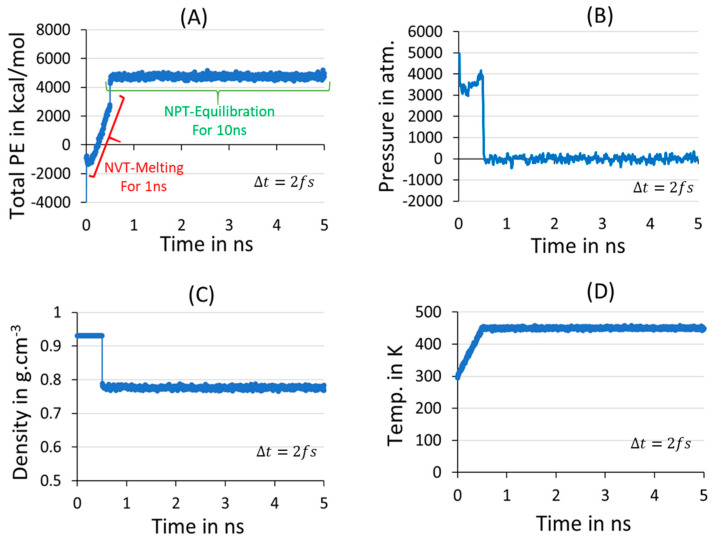
The isotropic mixture formation in the melting and equilibration step is accounted for 10C_1000_ model using potential energy (**A**), pressure (**B**), density (**C**), and temperature (**D**) for the NVT 1 ns melting and equilibration for 10 ns time length and NPT conditions of 450 K and 1 atm.

**Figure 3 polymers-16-01007-f003:**
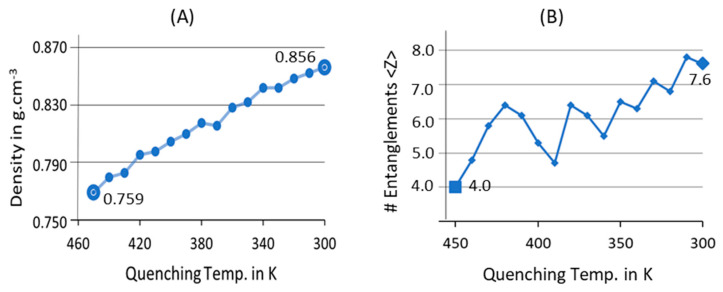
The density (**A**) and number of entanglements per chain (**B**) changes during the temperature-quenching simulation of 10C_1000_ model accounted for 450 K to 300 K at 1 atm and NPT conditions.

**Figure 4 polymers-16-01007-f004:**
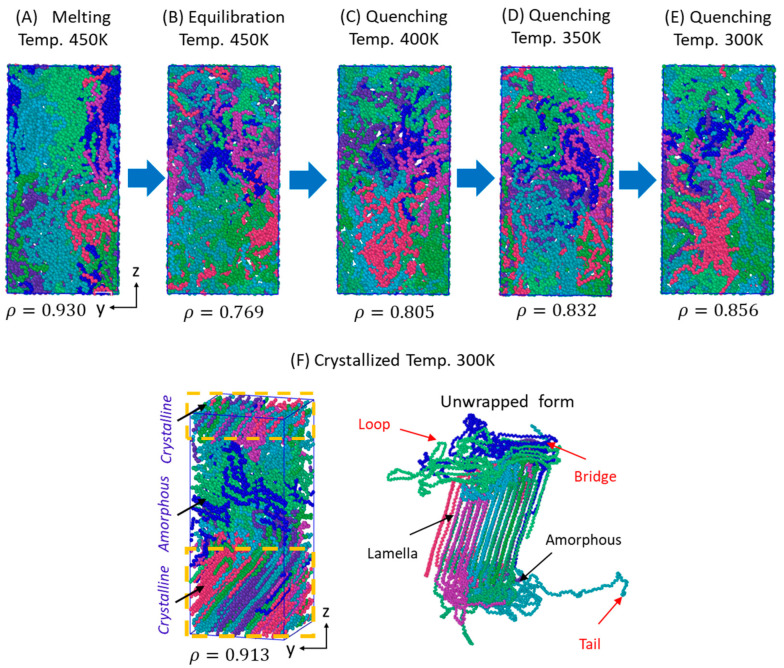
Structural changes during the MD simulations, starting from the LAMMPS data file to the isothermally crystallized model at 1µs for 10-chain models. Each chain represents a color in the studied model.

**Figure 5 polymers-16-01007-f005:**
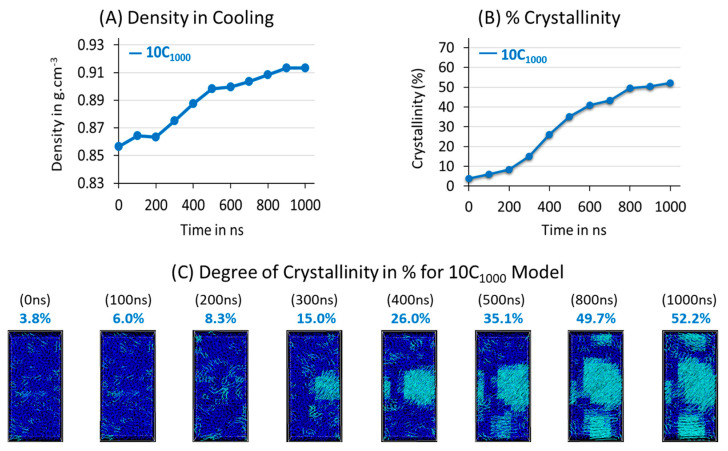
The computed total density (**A**) and degree of crystallinity (**B**,**C**) in isothermal crystallization at 300 K and 1 atm conditions are given for the 10C_1000_ model in blue. The $\chi_{c}$ is shown with the nanosecond simulation time (ns).

**Figure 6 polymers-16-01007-f006:**
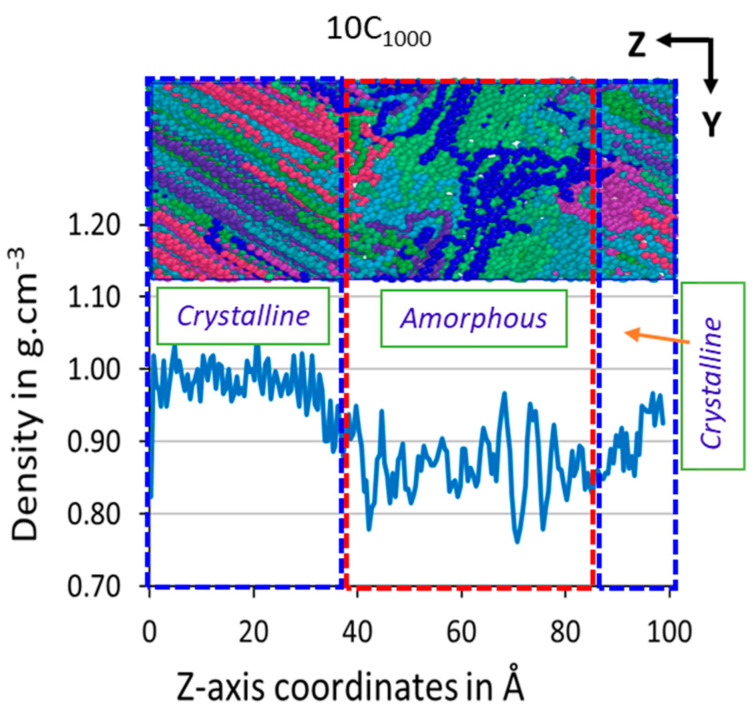
Z-axis dimensional density in the 10C_1000_ model evidences the formation of random-order polymer chains sandwiched between two highly oriented lamellar stems. The rectangular boxes represent the HDPE chain structures; each chain represents a unique color in both models. The graph with a density of ca. 1.0 g·cm^−3^ represents the crystalline phase (blue dotted region), and the amorphous state represents the density of ca. 0.85 g·cm^−3^ (red dotted lines).

**Figure 7 polymers-16-01007-f007:**
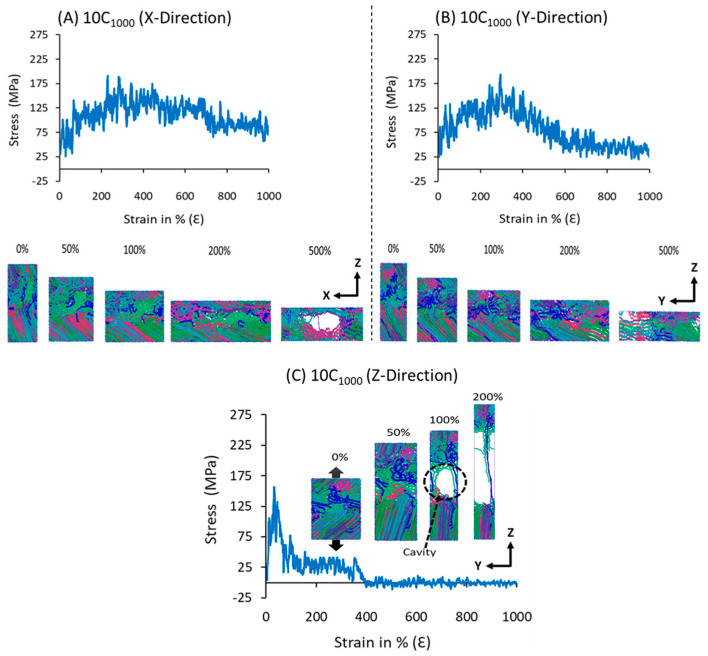
The S-S curve of the 10C_1000_ model is computed at 300 K and zero-pressure conditions using the NPT ensemble. The model is deformed to 1000% of the initial box length in all axis directions, aligning the lamellar stack orientation. The colors in the models represent each chain.

**Figure 8 polymers-16-01007-f008:**
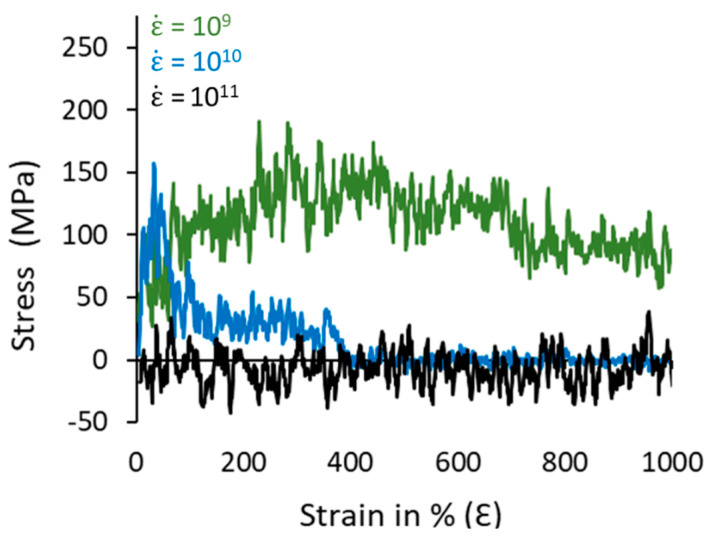
Strain-rate-dependent S-S curves for the 10C_1000_ model at 300 K and zero-pressure conditions using the NPT ensemble. Each model deformed to 1000% of the initial box length in the Z-axis direction, aligning the lamellar stack orientation.

**Table 1 polymers-16-01007-t001:** The PYS/R forcefield parameters employed to simulate the models using bonded and non-bonded terms are given in actual units.

Bonded Terms								Non-Bonded Terms		
Bond		Angle		Dihedral Angle				Lennar-Jones (12-6 Potential)		
*K* _b_	*r* _eq_	*K* _θ_	*θ* _eq_	*C* _0_	*C* _1_	*C* _2_	*C* _3_	*ε*	*σ*	Cut-off (*r*_cut_)
350.0	1.53	60.0	109.0	1.57	−4.05	0.86	6.48	0.112	4.01	9.0

## Data Availability

The data supporting this study’s findings are available upon request from the corresponding author.

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
