# Peer review of "Coarse-Grained Simulations on Polyethylene Crystal Network Formation and Microstructure Analysis"

_polymers, 2024, doi:10.3390/polym16071007_

Round 1

Reviewer 1 Report

Comments and Suggestions for Authors

1. The main issue addressed in this paper is related to the investigation of a model to predict the relationship between microstructure and mechanical properties of plastics under applied forces. The density distribution along the alternate direction of the plate orientation axis further confirms the orientational ordering, and tensile test simulations show that the stress-strain (SS) curve exhibits a reversible sharp elasticity regime, yield strength and irreversible plastic deformation regime, which are the characteristic SS curves for semi-crystalline polyethylene models. The cavitation mechanism involves the alignment of lamellar rods along the deformation axis and exhibits a Young's modulus of ca. 100 MPa. This study emphasises the formation of a crystalline network for the PE model using the PYS/R force field, and it produces a microstructure with ordered lamellar and amorphous segments with robust mechanical properties that are consistent with previous results.

2- The mathematical modelling technique is original, which allows a comprehensive study of the microstructure and mechanical properties of plastics under applied forces, which closes a gap in this research area.

3. The study presents an explanatory model of crystal network formation for PE model using PYS/R force field, and it produces a microstructure with ordered lamellar and amorphous segments with robust mechanical properties.

Refinements of the paper:

3.1)The abstract of the paper needs revision. Show numerical values and reflect the scientific novelty more clearly. Add quantitative results on the microstructure and mechanical properties of plastics.

3.2) The formulae underlying the numerical modelling should be added.

3.3) The change in crystallinity of the polymer should be justified analytically and a numerical evaluation should be introduced.

3.4) Improve the conclusions.

5. The conclusions should be supplemented by: Comparison of the numerical modelling results and their evaluation with third party data. 

6. References in the article are fully appropriate and allow to justify the direction of research development.

Comments on the Quality of English Language

Minor editing of English language required

Author Response

Comments and Suggestions: 1. The main issue addressed in this paper is related to the investigation of a model to predict the relationship between microstructure and mechanical properties of plastics under applied forces. The density distribution along the alternate direction of the plate orientation axis further confirms the orientational ordering, and tensile test simulations show that the stress-strain (SS) curve exhibits a reversible sharp elasticity regime, yield strength and irreversible plastic deformation regime, which are the characteristic SS curves for semi-crystalline polyethylene models. The cavitation mechanism involves the alignment of lamellar rods along the deformation axis and exhibits a Young's modulus of ca. 100 MPa. This study emphasizes the formation of a crystalline network for the PE model using the PYS/R force field, and it produces a microstructure with ordered lamellar and amorphous segments with robust mechanical properties that are consistent with previous results.

Comments and Suggestions: 2. The mathematical modelling technique is original, which allows a comprehensive study of the microstructure and mechanical properties of plastics under applied forces, which closes a gap in this research area.

Comments and Suggestions: 3. The study presents an explanatory model of crystal network formation for PE model using PYS/R force field, and it produces a microstructure with ordered lamellar and amorphous segments with robust mechanical properties.

Answer: The authors are sincerely thankful to the reviewers for their constructive suggestions on improving the manuscript with key details. The revised version includes the changes recommended by the reviewer, and the abstract and conclusions sections have been changed to strengthen the manuscript.

Comments on the refinements of the paper:

3.1) The abstract of the paper needs revision. Show numerical values and reflect the scientific novelty more clearly. Add quantitative results on the microstructure and mechanical properties of plastics.

Answer: The authors are sincerely thankful to the reviewers for their crucial suggestions in improving the manuscript. The new abstract version has the quantitative results for the mechanical properties computed for the HDPE plastic model and the crucial points are included.

3.2) The formulae underlying the numerical modeling should be added.

Answer: As per the reviewer's suggestion, the appropriate formulae and a few more equations for one-dimensional density and mechanical properties computations using MD simulations are included in the revised version of the manuscript.

3.3) The polymer's change in crystallinity should be justified analytically, and a numerical evaluation should be introduced.

Answer: The authors are thankful to the reviewer for his critical comments on crystallinity measurement in the submitted manuscript. However, the standard scientific protocol is followed to study the degree of crystallinity in the work. The numerical evaluation of the degree of crystallinity is clearly shown in Figure 5B, and Figure 5C visualizes it to justify its evolution with time in isothermal crystallization.

3.4) Improve the conclusions.

Answer: The revised version has the improved conclusions version and ensured that the reviewer’s suggested comments are also included in the revised version of the manuscript to strengthen the conclusion part. We appreciate the reviewer’s time and critical comments in this matter.

Comments and Suggestions: 4. The conclusions should be supplemented by: Comparison of the numerical modelling results and their evaluation with third party data. 

Answer: The revised version has included the changes as recommended by the reviewer.

Comments and Suggestions: 5. References in the article are fully appropriate and allow to justify the direction of research development.

Answer: We have accepted the reviewer’s comments and the appropriate changes are included in the revised version of the manuscript and the changes are highlighted in yellow to identify the changes.

Reviewer 2 Report

Comments and Suggestions for Authors

1)          One of my greatest concerns deals with the validation of simulated results. How to justify the accuracy of MD simulations? This is very crucial to me!

2)          Page 3, Line 127, the authors mentioned the NVT. Why not using the other ensemble?

3)          Page 4, how to justify the accuracy of the interactions and involved parameters?

4)          There are many grammar errors and badly worded/constructed sentences throughout the manuscript, promoting the poor readability in a few of parts. Language needs to be improved.

5)          The tense throughout the manuscript seems to be confused. Pls check it.

6)          About the citation of references, for example, "the two structural phases [1-5].", not ". [1-5]"

7)          Page 3 Line 106, {37,38}.

8)          Physical quantities should be italic. This is common sense.

9)          Equations need to be numbered.

Comments on the Quality of English Language

There are many grammar errors and badly worded/constructed sentences throughout the manuscript, promoting the poor readability in a few of parts. Language needs to be improved.

Author Response

Reviewer -2:

1)         One of my greatest concerns deals with the validation of simulated results. How to justify the accuracy of MD simulations? This is very crucial to me!

Answer: The authors understand the reviewer’s concern over the results obtained from the two methods used in this study. In the beginning, the single-chain models are allowed to be folded using the Material Studios’ Forcite module at the arbitrary parameters and ensemble (NVT) assigned in the simulations. This method is to obtain a lamellar folded chain conformation for the HDPE model.

The main MD simulations of HDPE are performed using the PYS/R forcefield parameters after preparing a rectangular cell box with the folded chain conformations. The validity of the models is discussed in terms of the temperature-dependent densities and validated with the previous publications. For example, Polymers 2022, 14, 5144. Additionally, the simulated density values are correlated with the appropriate references available in the literature.

  1. The density: The density of the molten state at 450K in the simulation model is 0.760 g.cm-3, which is close to the experimental value reported for PE of 0.765 g.cm-3 [Ref: Pearson, D. S.; Ver Strate, G.; von Meerwall, E.; Schilling, F. C. Macromolecules 1987, 20, 1133]. Similarly, theoretical references appear with a similar range of density in the articles of Moorthi, K.; Kamio, K.; Ramaos, J.; Theodorou, D. N. Macromolecules 2012, 45, 8453; Ramaos, J.; Vega, J.F.; Marinez-Salazar, J. European Polymer Journal 2018, 99, 298-331. More recently, the density of 0.766 g.cm-3 is reported by Sefiddashti et al in Polymers 2023, 15, 1831.]
  2. Density in the amorphous state at 300K: The reported density at 300K after the temperature quenching from the melt is 0.859 g.cm-3, which is also closer to the reported value of 0.85 g.cm-3 in the HDPE simulated model density of 0.850 g.cm-3 by Mohammad Atif et al. ACS Appl. Polym. Mater. 2021, 3, 2, 620–630.
  3. As per the semicrystalline model obtained in the simulations at 300K and 1 atm, the density is 0.911 g.cm-3, slightly underestimated compared to the experimental density of 0.94-0.97 g.cm-3. Such underestimation of the densities is also reported for the HDPE model simulations. [Ref.64]
  4. In addition, the production run - isothermal crystallization results are based on the equilibration done at the melt state at 450K. So, the pressure, density, and potential energy parameters are also reported in figure 2 of the manuscript, which is evident that that the steady state is reached in the time for equilibration, 10 ns, using the NPT ensemble, suggesting the obtained results are completely relaxed at 450K.

2)         Page 3, Line 127, the authors mentioned the NVT. Why not using the other ensemble?

Answer: The polymer chain is simulated in isolated form using Material Studio’s Forcite module. The simulation aimed to obtain a folded lamellar chain conformation in a short run, which would be used to pack them in a rectangular box for the main simulations in the LAMMPS code and standard PYS/R forcefield parameters. The NVT ensemble is fast and flexible for our purpose. As no forcefields are available for the united atom model simulations in Materials Studio, the universal forcefield parameters mimicked the CH2 atom as only C-atom. The choice of the ensemble is arbitrary rather than guided by specific rules or physical principles. The main focus of the authors is to provide and discuss the results from reliable forcefield parameters such as the PYS/R for HDPE models, while taking a starting structure with folded conformation. The aim of the simulations is to obtain a semi-crystalline model with the least computational efficiency, which is quite successful in our study.

3)          Page 4, how to justify the accuracy of the interactions and involved parameters?

Answer: The PYS/R forcefield is a well-known forcefield for the HDPE models, as evidenced in the literature. The interactions obtained would be reliable and consistent with previous studies. The manuscript has described all the interactions and parameters related to concerns in Figure 2 and Figure S2.

4)         There are many grammar errors and badly worded/constructed sentences throughout the manuscript, promoting the poor readability in a few of parts. Language needs to be improved.

Answer: The revised manuscript is modified with the recommended changes and highlighted in yellow.

5)          The tense throughout the manuscript seems to be confused. Pls check it.

Answer: The revised manuscript is modified with the recommended changes and highlighted in yellow.

6)          About the citation of references, for example, "the two structural phases [1-5].", not ". [1-5]"

Answer: The revised manuscript is modified with the recommended changes and highlighted in yellow.

7)         Page 3 Line 106, {37,38}.

Answer: The revised manuscript is modified with the recommended changes and highlighted in yellow.

8)          Physical quantities should be italicized. This is common sense.

Answer: The revised manuscript is modified with the recommended changes.

9)          Equations need to be numbered.

Answer: The revised manuscript is modified with the recommended changes and highlighted in yellow.

In addition, a few more equations are introduced at the necessary places in the manuscript.

Round 2

Reviewer 1 Report

Comments and Suggestions for Authors

Accept in present form

Reviewer 2 Report

Comments and Suggestions for Authors

The comments have been addressed appropriately in the revised manuscript. Therefore, the manuscript is suitable for publication in POLYMERS.